# Analysis of Polymorphism rs1333049 (Located at 9P21.3) in the White Population of Western Siberia and Associations with Clinical and Biochemical Markers

**DOI:** 10.3390/biom9070290

**Published:** 2019-07-19

**Authors:** Elena Shakhtshneider, Pavel Orlov, Sergey Semaev, Dinara Ivanoshchuk, Sofia Malyutina, Valery Gafarov, Yuliya Ragino, Mikhail Voevoda

**Affiliations:** 1Institute of Internal and Preventive Medicine—branch of Institute of Cytology and Genetics, Siberian Branch of Russian Academy of Sciences (SB RAS), Bogatkova Str. 175/1, Novosibirsk 630004, Russia; 2Federal research center Institute of Cytology and Genetics, SB RAS, Prospekt Lavrentyeva 10, Novosibirsk 630090, Russia

**Keywords:** rs1333049, risk of cardiovascular disease, lipid profile, white population, Western Siberia

## Abstract

The 9p21.3 chromosomal region is a marker of the risk of cardiovascular diseases. The aim of this study was to analyze single-nucleotide polymorphism rs1333049 (chr9:22125504) in the population of Western Siberia (Russia) and possible associations with clinical and biochemical parameters. The population included in the analyses was selected from a sample surveyed within the framework of the Health, Alcohol and Psychosocial Factors In Eastern Europe (HAPIEE) study (9360 participants, >90% white, aged 45–69, males: 50%). In total, 2729 randomly selected patients were included. Plasma lipid levels were determined by standard enzymatic assays. Rs1333049 was analyzed by RT-PCR (BioLabMix, Russia). Frequencies of rs1333049 genotypes C/C (homozygote), C/G (heterozygote), and G/G were 0.22, 0.51, and 0.27 in this population. The Allele G frequency was 0.53. We found an association of allele G with total cholesterol and low-density lipoprotein cholesterol levels among male participants (*p* = 0.004 and *p* = 0.002, respectively). Allele C was significantly associated with the risk of myocardial infarction among the male participants (odds ratio 1.96, 95% confidence interval 1.14–3.38, *p* =  0.017) and the study population (odds ratio 1.83, 95% confidence interval 1.23–2.72, *p * =  0.004). Thus, rs1333049 is associated with myocardial infarction in the white population of Western Siberia (Russia).

## 1. Introduction

One of the markers of cardiovascular disease (CVD) risk is chromosomal region 9p21.3. It contains two genes (encoding proteins CDKN2A and CDKN2B) and the gene of long noncoding RNA ANRIL (in the antisense orientation at the *INK4* locus). It has been shown that ANRIL regulates *CDKN2A* and *CDKN2B* expression. The two resultant proteins, CDKN2A and CDKN2B (also known as p15^INK4B^ and p16^INK4A^) are inhibitors of a cyclin-dependent kinase (CDK4 and CDK6, respectively), whereas the p14^ARF^ protein is expressed from an alternative frame of *CDKN2B* (Figure 1). These proteins participate in the regulation of the two main tumor-suppressor pathways: RB and p53 (also known as TP53) cascades. Single-nucleotide polymorphisms (SNPs) in this region (rs10811656, rs10757278, and rs1333049) are related to the regulation of ANRIL splicing products and/or *ANRIL* expression. For instance, the double-carrier status on the minor alleles of rs10811656 (T) and rs10757278 (G) disrupts the STAT1-binding site and increases *ANRIL* expression [1,2]. 

Population studies on rs1333049 have uncovered associations with CVDs and type 2 diabetes [3,4,5,6,7,8,9]. Patel et al. did not find a significant association between chromosome 9p21 and death/myocardial infarction as the primary outcome of CHD among patients with established CHD at baseline (odds ratio, 1.02; 95% CI, 0.99–1.05) in a meta-analysis. In contrast to studies comparing individuals with CHD to disease-free controls, they found no clear association between genetic variation in chromosomal region 9p21 and the risk of subsequent acute CHD events when all individuals had CHD at baseline. Nonetheless, they uncovered an association with subsequent revascularization, which may support the postulated mechanism of promotion of atheroma development by chromosomal region 9p21 [10]. Data from Russian patients have confirmed the correlation of rs1333049 with myocardial infarction but not with type 2 diabetes; these studies had some limitations because they involved only small groups [11,12]. The specificity of rs1333049 as a risk factor of CVDs in various populations and ethnic groups makes this topic relevant for the populations living under various geographic/climatic conditions, including the residents of Western Siberia (Russia). 

Our aim was to analyze the association of rs1333049 with clinical and biochemical parameters in the population of Western Siberia.

## 2. Materials and Methods

A cross-sectional epidemiological study on an adult population was conducted in Novosibirsk (Western Siberia, Russia). The main representative sample (9360 people, age 45–69 years, 53.8 ± 7.0 [mean ± standard error]) was compiled by means of a random number table of Novosibirsk residents analyzed in 2007–2008 during the screening within the framework of the Health, Alcohol and Psychosocial Factors In Eastern Europe (HAPIEE) study (supervised from the Wellcome Trust Foundation, Great Britain) [13]. The vast majority of the analyzed population consisted of whites (>90%). The study protocol was approved by the Ethics Committee of the Institute of Internal and Preventive Medicine (a branch of the Institute of Cytology and Genetics, SB RAS). Written informed consent was obtained from each participant regarding study participation, the medical examination, and collection and analysis of biological materials including genetic tests. 

The analyses included the recording of social-demographic data, a medical examination, a standard questionnaire about smoking, anthropometric measurements (height, body weight, and waist circumference), measurement of arterial blood pressure, and biochemical analyses of blood serum [total cholesterol (TC), high-density lipoprotein cholesterol (HDL-C), low-density lipoprotein cholesterol (LDL-C), triglycerides (TGs), and fasting blood serum glucose].

Blood collection from the median cubital vein was performed in the morning after an overnight fast for 12 h. The blood lipid indicators (TC, TGs, HDL-C, and LDL-C) were measured by enzymatic methods with standards from Biocon Fluitest (Lichtenfels, Germany) on a Labsystem FP-901 biochemical analyzer (Helsinki, Finland) according to the manufacturer’s instruction. The atherogenic index (AI) was calculated via the following formula: AI = (TC – HDL-C)/HDL-C.

For a molecular-genetic analysis, 2729 subjects were selected from the study population (*N* = 9360) by the random number method. DNA was isolated from blood by phenol:chloroform extraction [14]. 

Rs1333049 was genotyped by the commercial KASP assay [15] designed by Biolabmix, Russia and the HS-qPCR Hi-ROX (2×) (BioLabMix, Novosibirsk, Russia) on a StepOnePlus Real-Time PCR System (Thermo Fisher Scientific, Foster City, CA, USA); 1 μL of assay mix, 10 μL of qPCR Hi-ROX (2×) master mix, 1 μL of DNA and 8 μL of water with a total final volume of 20 μL were used. The oligonucleotides used for rs1333049 genotyping: specific allele C–5′gagggtgaccaagttcatgcttaaccatatgatcaacagttc3′; specific allele G–5′gagggtgaccaagttcatgcttaaccatatgatcaacagttg3′; common-atttacatttccttcactactg; hydrolysis probe assays VIC aggacgctgagatgcgtcct*gaaggtcggagtcaacggatt; hydrolysis probe assays FAM agcgatgcgttcgagcatcgct*gagggtgaccaagttcatgct. The thermocycling programs consisted of initial denaturation at 95 °C for 5 min, and then 20 cycles at 95 °C for 10 s, annealing temperature at 54 °C for 10 s and 72 °C for 10 s, 30 cycles at 95 °C for 15 s, annealing temperature at 52 °C for 10 s and extension 72 °C for 10 s (detection) with post PCR read 30 s at 60 °C. Laboratory personnel performing genotyping assays were blinded to physical and clinical examination.

For statistical analysis, the significance of differences in allele frequencies among subgroups and analysis of compliance with the Hardy–Weinberg equilibrium were carried out by the χ^2^ test. Differences in means of continuous variables among the genotypes were evaluated after adjustment for sex, age, and the body–mass index according to the GLM model of the SPSS software suite for Windows. 

## 3. Results 

Baseline characteristics of the subjects are presented in Table 1. The males constituted 46.5% and females 53.5%. The prevalence of hypertension (>140/90 mm Hg) was 42.3%, type 2 diabetes 7.9%, dyslipidemia (TC > 200 mg/dL, or 5.2 mmol/L) 82.8% of all subjects.

The frequencies of rs1333049 alleles in the white population of Western Siberia are listed in Table 2.

The associations of rs1333049 with blood lipid indicators—TC, HDL-C, LDL-C, TGs, and the atherogenic index—were studied next. Differences among the genotypes in mean levels of TC in blood were statistically significant (*p* = 0.004) only among males (Table 3): Higher mean levels of TC were noted for the genotypes involving the G allele. Statistically significant differences in mean TC among the genotypes were not detected among the females and in the total study population (*p* > 0.05).

Differences among rs1333049 genotypes in mean blood levels of LDL-C were also statistically significant (*p* = 0.002) only in males (Table 3): Higher mean levels of LDL-C in this group corresponded to genotypes G/G and C/G. Statistically significant differences in mean LDL-C among the genotypes were not detected among the females and in the total study population (*p* > 0.05).

According to the GLM, in the analyzed groups, there were no statistically significant correlations of rs1333049 with HDL-C, TGs, the atherogenic index, blood glucose, body–mass index, systolic blood pressure, diastolic blood pressure, and heart rate.

Thus, we uncovered a statistically significant association of rs1333049 genotypes with TC and LDL-C among the white males of Western Siberia. 

For 10 years (2007–2017), in the study population, investigators collected the data on new cases of myocardial infarction on the basis of the Novosibirsk City Registry of Myocardial Infarction [16]. In total, during the study period, 509 new cases of myocardial infarction were registered. In the subsample of 2729 subjects, which was genotyped for rs1333049, there were 118 new cases of myocardial infarction. In the white population of Western Siberia, we confirmed the association of rs1333049 with myocardial infarction among males (Table 4). Therefore, carriage of the C allele is a risk factor of myocardial infarction.

## 4. Discussion

The frequencies of rs1333049 alleles in the white population of Western Siberia were found to be consistent with those of various European populations: according to the data from gnomAD—Genomes European, G = 0.528, C = 0.472 (http://www.ncbi.nlm.nih.gov).

For SNP rs1333049 in the total study population and among females, the frequencies of genotypes were compliant with the Hardy–Weinberg equilibrium. In males, the observed distribution of genotype frequencies did not match the theoretically expected one because of lower frequencies of genotypes G/G (homozygous) and C/G (heterozygous). The observed noncompliance with the Hardy–Weinberg equilibrium among the rs1333049 genotype frequencies in males most likely indicates selection for this SNP.

Carriage of the C allele is a risk factor of myocardial infarction in the white population of Western Siberia. Such an association has been found in various ethnic groups elsewhere. The minor allele (risk allele) of rs1333049 (C) is widespread across the globe. This allele raises the risk of ischemic heart disease by 15–20% in the heterozygous state and by 30–40% in the homozygous state [17,18,19]. According to the literature, chromosomal locus 9.21, where rs1333049 is located, may be involved in the signaling pathway associated with inflammation in the arterial wall [20].

Individual risk of CVDs depends on both genetic factors and lifestyle factors. This study presents the results of our analysis of a Russian population for the association of SNP rs1333049 with clinical and biochemical parameters. Changes in DNA sequence are an independent risk factor of myocardial infarction, in agreement with the findings of other authors [21,22,23,24,25,26,27,28,29,30,31,32]. The risk of an unfavorable outcome depends on the presence of a certain allele or genotype. 

Our study revealed an association of the G allele of rs1333049 with TC and LDL-C levels among males. These data contradict some other studies [10,11,33], for example, Ellis et al. have demonstrated a correlation of the C/C genotype with higher TG and TC levels in blood [26]. The study by Ellis et al. was performed on patients with ischemic heart disease (New Zealand), whereas our study deals with the general population. Perhaps the observed association may be explained by clinical and group features of the populations being analyzed. In most studies on groups of patients, an association of rs1333049 with lipid metabolism parameters has not been detected [34]. 

Our study confirmed the association of rs1333049 with myocardial infarction in the general population; this finding is consistent with the results of studies on some groups of patients in Russia [11,12,35] and abroad [35,36,37,38]. 

According to the GLM, in the analyzed groups, there were no statistically significant correlations of rs1333049 with blood glucose levels. In other studies, there have been no statistically significant correlations of rs1333049 with type 2 diabetes in Russia [11,12]. Here, in 242 patients out of 2729, the diagnosis was type 2 diabetes. According to the GLM, in the analyzed groups, there were no statistically significant correlations of rs1333049 with type 2 diabetes in the white population of Western Siberia.

Our study has some limitations. We analyzed only rs1333049 (which is located in the 9p21.3 region) and traditional CVD risk factors; therefore, we could not rule out the influence of other factors that may affect the results of observational studies. 

The research into genetic risk factors of CVDs is important not only for the analysis of disease outcomes but also for preventive measures, considering that genetic variants can be detected before the first clinical manifestations of a CVD. Accordingly, patients at a high risk of CVD (in terms of genetic factors) may have additional motivation to lead a healthy lifestyle [21]. Furthermore, information about genetic risk factors of a disease may be employed to improve the clinical management of the patients.

## Figures and Tables

**Figure 1 biomolecules-09-00290-f001:**
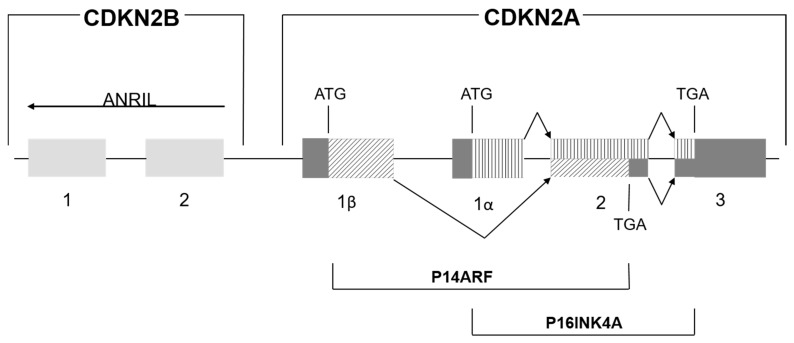
The *CDKN2B/CDKN2A/ANRIL* locus in chromosomal region 9p21.

**Table 1 biomolecules-09-00290-t001:** Baseline characteristics of the subjects.

	Males	Females	Both Sexes
Number of subjects	1270	1459	2729
Age, years	56.7 ± 0.2	56.6 ± 0.2	56.7 ± 0.1
TC, mg/dL	241.5 ± 1.4	258.8 ± 1.5	250.8 ± 1.1
HDL-C, mg/dL	58.2 ± 0.4	61.4 ± 0.5	59.9 ± 0.3
LDL-C, mg/dL	119.1 ± 1.3	131.8 ± 1.3	125.9 ± 0.9
TGs, mg/dL	141.5 ± 2.2	144.4 ± 2.2	143.1 ± 1.6
Index of atherogenicity	2.9 ± 0.04	3.0 ± 0.04	2.9 ± 0.03
Fasting glucose, mmol/L	5.8 ± 0.1	5.8 ± 0.1	5.8 ± 0.1
Body mass index, kg/m^2^	26.7 ± 0.1	29.8 ± 0.2	28.3 ± 0.1
Waist circumference, cm	95.4 ± 0.4	92.3 ± 0.4	93.8 ± 0.3
Systolic blood pressure, mmHg	143.5 ± 0.7	143.8 ± 0.7	143.7 ± 0.5
Diastolic blood pressure, mmHg	90.4 ± 0.4	89.9 ± 0.4	89.9 ± 0.3
Heart rate, bpm	72.3 ± 0.4	72.0 ± 0.3	72.1 ± 0.2

* Continuous variables are presented as mean ± standard error. TGs, triglycerides; TC, total cholesterol; HDL-C, high-density lipoprotein cholesterol; LDL-C, low-density lipoprotein cholesterol.

**Table 2 biomolecules-09-00290-t002:** Frequencies of the alleles and homozygous and heterozygous genotypes of rs1333049.

	**Males**	**Females**	**Both Sexes**
	**%** *n* = 1270	**%** *n* = 1459	**%** *n* = 2729
**Genotypes**			
C/C	0.21 *n* = 271	0.22 *n* = 322	0.22 *n* = 593
C/G	0.53 *n* = 670	0.49 *n* = 712	0.51 *n* = 1382
G/G	0.26 *n* = 329	0.29 *n* = 425	0.27 *n* = 754
**Alleles**			
C	0.48	0.465	0.47
G	0.52	0.535	0.53
**Indicator of compliance with Hardy–Weinberg equilibrium**			
χ^2^	4.17	0.53	0.73

*n*: number of subjects, χ^2^: chi-squared.

**Table 3 biomolecules-09-00290-t003:** Associations of rs1333049 genotypes with biochemical parameters in the white population of Western Siberia.

**Sex**	**Genotype**	**TC, mg/dL**	**HDL-C, mg/dL**	**LDL-C, mg/dL**	**TGs, mg/dL**	**Atherogenic index**	**Fasting glucose, mmol/L**	**Body–mass index, kg/m^2^**	**Systolic blood pressure** **, mmHg**	**Diastolic blood pressure** **, mmHg**	**Heart rate, bpm**
**Male**	C/C	232.9 ± 3.1	58.1 ± 0.9	111.3 ± 2.7	140.0 ± 4.8	2.77 ± 0.08	5.57 ± 0.14	26.7 ± 0.3	143.5 ± 1.4	90.6 ± 0.8	71.9 ± 0.8
	C/G	242.5 ± 1.9	58.2 ± 0.6	119.7 ± 1.7	142.8 ± 3.1	2.90 ± 0.05	5.87 ± 0.09	26.4 ± 0.2	143.0 ± 0.9	90.1 ± 0.5	72.0 ± 0.5
	G/G	246.3 ± 2.8	58.4 ± 0.9	124.4 ± 2.5	140.2 ± 4.4	2.96 ± 0.08	5.77 ± 0.12	27.2 ± 0.3	144.5 ± 1.2	90.7 ± 0.7	73.3 ± 0.7
***p***		**0.004** *	0.954	**0.002** *	0.835	0.229	0.190	0.061	0.620	0.785	0.296
**Female**	C/C	265.1 ± 3.1	61.8 ± 1.1	133.5 ± 2.8	152.6 ± 4.6	3.00 ± 0.08	6.00 ± 0.14	29.9 ± 0.3	144.1 ± 1.3	90.4 ± 0.7	72.8
	C/G	257.1 ± 2.1	61.5 ± 0.7	130.9 ± 1.9	142.3 ± 3.1	2.95 ± 0.06	5.66 ± 0.09	29.7 ± 0.2	143.2 ± 0.9	89.4 ± 0.5	72.3
	G/G	257.1 ± 2.7	60.9 ± 0.9	131.9 ± 2.4	141.9 ± 4.0	2.92 ± 0.07	5.70 ± 0.12	29.7 ± 0.3	144.6 ± 1.2	90.4 ± 0.6	71.9
***p***		0.080	0.787	0.302	0.133	0.744	0.116	0.811	0.615	0.404	0.605
**Both sexes**	C/C	250.3 ± 2.2	60.0 ± 0.7	123.3 ± 1.9	147.0 ± 3.4	2.90 ± 0.06	5.81 ± 0.10	28.4 ± 0.2	143.9 ± 1.0	90.5 ± 0.6	72.4 ± 0.5
	C/G	250.4 ± 1.5	59.9 ± 0.5	125.7 ± 1.3	142.6 ± 2.2	2.93 ± 0.04	5.76 ± 0.06	28.2 ± 0.1	143.1 ± 0.6	89.8 ± 0.4	72.1 ± 0.3
	G/G	251.8 ± 1.9	59.7 ± 0.6	128.2 ± 1.7	140.9 ± 2.9	2.94 ± 0.05	5.73 ± 0.09	28.5 ± 0.2	144.5 ± 0.9	90.5 ± 0.5	72.6 ± 0.5
**p**		0.825	0.937	0.173	0.386	0.896	0.835	0.350	0.397	0.348	0.770

The data are presented as mean ± standard error. *A statistically significant difference among genotypes.

**Table 4 biomolecules-09-00290-t004:** Association of rs1333049 with myocardial infarction in the white population of Western Siberia.

Sex	Genotype	Population		Myocardial Infarction		OR(95% CI)	P
		n	%	n	%		
Male	C/C	250	20.7	21	33.9	1.96(1.14–3.38)	**0.017** *
	C/G	644	53.3	26	41.9	0.63(0.38–1.06)	0.09
	G/G	314	26	15	24.2	0.91(0.5–1.65)	0.882
Female	C/C	302	21.6	18	32.1	1.72(0.97–3.06)	0.07
	C/G	687	49	25	44.6	0.84(0.49–1.43)	0.589
	G/G	412	29.4	13	23.2	0.73(0.39–1.36)	0.37
Both sexes	C/C	554	21.2	39	33.1	1.83(1.23–2.72)	**0.004** *
	C/G	1330	51	51	43.2	0.73(0.51–1.06)	0.11
	G/G	726	27.8	28	23.7	0.81(0.52–1.24)	0.4

* Statistical significance.

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
