# Peer review of "Analysis of Polymorphism rs1333049 (Located at 9P21.3) in the White Population of Western Siberia and Associations with Clinical and Biochemical Markers"

_biomolecules, 2019, doi:10.3390/biom9070290_

Round 1
Reviewer 1 Report
This is a study about the association between polymorphism rs1333049 and markers related to myocardial infarction. The design and the results obtained are correct but the impact of this analysis is questionable since data comes from a specific local population and the main conclusion (lines 30-31: Rs1333049 is associated with myocardial infraction…) has been widely probed in multiple populations as the authors admit several times in the manuscript.
Other minor points:
-2729 patients were finally selected. Reasons for this number of patients must be specified.
-Line 63: Dr. SK Malyutina is a coauthor of the manuscript. Therefore, he /she should not be cited in Materials and Methods as the source of the data.
-Table 1 is not cited in the text.
-There are several paragraphs of the results that should be moved to the Discussion section: lines 92-94, lines 100-102, lines 131-135.
Author Response
Thank you for the thorough review of our paper.
Cardiovascular diseases take the leading position within the mortality structure of the adult population in the economically developed countries around the world. In Russia, according to data provided by the Federal State Statistics Service, the cardiovascular mortality rates in 2016 were 616.4 cases per 100,000 of the population, which is several times higher than in the developed countries of the world. At the same time, it was shown that there are many problems in calculating the impact of specific gene polymorphism to the development and manifestation of the pathological process. In some cases, the assessment of the gene impact is complicated by the existence of distinct haplotypes in ethnic groups.
- The size of the final sample of 2729 patients was due to size required to confidently observe an anticipated effect.
- We corrected this item in the Materials and Methods section, line 63.
- We added the citation of Table 1 to the text.
- We changed the text and moved the specified paragraphs to the Discussion section, lines 147-161.
Reviewer 2 Report
I read with great interest the article by Shakhtshneider et al on the role of SNP rs1333049 in thepopulation of Western Siberia. I have a few minor revision to made:
1: Written informed consent was obtained from each participant regarding also genetic test? It should be specify in the methods.
2: Add Table 1 in the results as population characteristics, before genetic results.
3: In the Table 1 please explain the "gender" row that is not clear (Males, % : Male 46.5 Female 53.5 Both sexes 46.5)
4: Since during the study period, new cases of myocardial infarction were registered, I would suggest to add the description of cardiovascular risk factors in the population: hypertension, diabetes, dyslipidemia and so on, since Table 1 does not clarify it.
5 : Please add in the Table 2 the numerosity of Male and Female groups
6: Since Authors describe only level of fasting glucose it is not clear if any correlation of SNP rs1333049 with presence of diabetes in this population is present.
7. In the context of genetgic variants and susceptiblity for ischemic heart disease, consider the following articles:
Cell J. 2015 Spring;17(1):89-98. Epub 2015 Apr 8.
Biomolecules. 2018 Dec 5;8(4). pii: E164. doi: 10.3390/biom8040164
Int. J. Mol. Sci. 2018, 19, 802; doi:10.3390/ijms19030802
J Diabetes Res. 2019 Apr 4;2019:9489826. doi: 10.1155/2019/9489826
Author Response
Thank you for your time and interest in our work
1: Written informed consent to conduct a genetic test was obtained from each participant. We added this information in the text, line 74.
2: We moved table 1 to the results section.
3: We corrected table 1. The data on the percentage of men and women participating in the study were indicated in the text before table 1, lines 107-108.
4: We added the information: lines 108-109.
5: We added in Table 2 the number of men and women in groups.
6: We did not reveal the association of rs1333049 with SD2 in the population of Western Siberia, lines 177-181.
7: We added links to the Discussion section, lines 288-296.
Reviewer 3 Report
Introduction- More background information and proper references needs to be provided.
Materials and methods- Not sufficient details provided for reproducibility of the study.
For example in line 84-85-
DNA was isolated from blood by phenol:chloroform extraction.Reference missing for phenol:chloroform extraction protocol.
SNP rs1333049 was analyzed by RT-PCR (Biolabmix, Russia). Details and sequence of the primer used for RT-PCR are missing.
The blood lipid indicators (TC, TGs, HDL-C, and LDL-C) were measured by enzymatic methods with standards from Biocon Fluitest (Germany) on a Labsystem FP-901
biochemical analyzer (Finland). What protocol was used? Did they performed all the assays according to the manufacturer's instruction, details need to be provided.
Not much details about analysis is provided, we also need to know was the analysis done by blinded observer? If yes it needs to be mentioned.
Results and discussion- More elaborated explanation needs to be provided for their results to make a strong claim about the association of rs1333049 with myocardial infarction in the general population - for example- Our study revealed an association of the G allele of rs1333049 with TC and LDL-C levels among males. These data contradict some other studies [10, 26], for example, Ellis et al. have demonstrated a correlation of the С/С genotype with higher TG and TC levels in blood [26]. Despite of the contradictory results form earlier published studies, just giving an explanation of using different study population is not sufficient enough. This discussion needs to be elaborated more.
Author Response
Thank you for perusing our manuscript
We have added to the Introduction section the lines 50-58.
We added to the section Materials and methods:
- lines 86-87
- lines 88-99
- line 83
- lines 99-100
We have added to the discussion section the lines 147-161 and 177-181.
Round 2
Reviewer 1 Report
Authors adequately answered questions raised by the reviewer.